# The Combined Effects of Dietary Diversity and Frailty on Mortality in Older Taiwanese People

**DOI:** 10.3390/nu14183825

**Published:** 2022-09-16

**Authors:** Wei-Ching Huang, Yi-Chen Huang, Meei-Shyuan Lee, Jia-Yau Doong, Wen-Harn Pan, Hsing-Yi Chang

**Affiliations:** 1Graduate Institute of Life Sciences, National Defense Medical Center, Taipei 114201, Taiwan; 2Institute of Population Health Sciences, National Health Research Institutes, Miaoli 35053, Taiwan; 3Department of Nutrition, China Medical University, Taichung 406040, Taiwan; 4School of Public Health, National Defense Medical Center, Taipei 114201, Taiwan; 5Department of Nutritional Science, Fu Jen Catholic University, New Taipei City 242062, Taiwan; 6Institute of Biomedical Sciences, Academia Sinica, Taipei 11529, Taiwan

**Keywords:** frailty, dietary diversity, mortality, older adults, Nutrition and Health Survey in Taiwan (NAHSIT)

## Abstract

Objective: To assess the prospective association between frailty and dietary diversity on mortality. Method: This prospective cohort study used the 2005–2008 Nutrition and Health Survey in Taiwan (N = 330; age ≥ 65 years) and this was linked to the Death Registry where we used the data that was recorded up to 31 January 2020. Dietary intake information was assessed using a 24-h dietary recall and food-frequency questionnaire, which were calculated a dietary diversity score (DDS; range, 0–6) and food consumption frequency. Assessment of frailty phenotypes was based on FRAIL scale which was proposed by the International Academy on Nutrition and Aging. Results: Frail older adults had a higher risk of all-cause mortality when they were compared to those with robust physiologies (hazard ratio [HR]: 3.73, 95% confidence interval [CI]: 2.13–6.52). Frailty and a lower DDS were associated with a higher risk of mortality (joint adjusted HR: 2.30, 95% CI: 1.11–4.75) which, compared with a robust physiology and higher DDS, were associated with a lower risk of mortality. Conclusions: Frailty and a lower DDS were associated with a higher mortality. Prefrailty and frailty with a higher DDS were associated with a lower risk of mortality when compared with those with prefrailty and frailty and a lower DDS. These results suggest that eating a wide variety of foods might reduce the risk of mortality in older adults with prefrailty and frailty.

## 1. Introduction

Frailty is a medical and complicated geriatric syndrome that is characterized by multisystem decline that is associated with decreased functional reserves and increased vulnerability with aging [1]. Most studies define physical frailty by unintentional weight loss, exhaustion, low physical activity, slowness, and low grip strength [2]. The possible causes of frailty include physiological, genetic, environmental, and nutritional factors [1]. The results from a systematic review and meta-analysis reported that the prevalence of frailty and prefrailty increases with age and it appears higher prevalence in upper middle-income countries, among community-dwelling older adults [3]. With the rapid growth of the older population, worldwide, frailty requires attention because those with this condition are at a higher risk of falling, developing a disability, hospitalization, and mortality, and they require more medical services [4].

Eating a healthy diet is an important non-pharmacological strategy for not only preventing and improving the variety of non-communicable diseases but also promoting independence, quality of life, and ultimately healthy aging [5,6]. Studies have explored dietary patterns [7,8] and dietary quality and how they might affect health or mortality [9]. A brief dietary measurement method that can predict general health or mortality would be useful for public health nutrition programs and in a community-dwelling setting. The dietary diversity score (DDS) is a dietary quality measurement method, which is a simple and rapid tool that requires no devices or complicated measurements [9,10]. Older adults with more dietary diversity usually have a better dietary nutrient and energy intake profile as well as a lower mortality [9].

Our previous study showed that older adults with frailty had a lower dietary diversity, and those with both had a higher risk of developing worsening cognitive function [11]. A Taiwanese cross-sectional study demonstrated that a dietary pattern with more fruit, nuts and seeds, tea, vegetables, whole grains, shellfish, milk/yogurt/cheese, and fish was associated with a reduced prevalence of frailty [12]. It is possible that the cause of frailty might not just be due to the inadequate intake of individual nutrients or foods. Older adults with a higher dietary diversity may have a reduced mortality that is related to cognitive impairment [13], poor appetite [14], and diabetes mellitus [15]. Moreover, the results from a prospective cohort study demonstrated that a poor diet quality may increase the incidence of frailty [16], subsequently leading to a higher risk of mortality [17]. 

It is unknown whether dietary diversity can reduce the mortality that is associated with frailty. Therefore, the aim of this study was to investigate the prospective associations of dietary diversity and frailty severity to all-cause mortality in older adults.

## 2. Materials and Methods

### 2.1. Participants’ Data Sources

The data were obtained from the Nutrition and Health Survey in Taiwan (NAHSIT) 2005–2008; detailed information on the design and methods of this survey has been published elsewhere [18]. A total of 968 older adults (aged ≥ 65 years) completed the household interviews. There were 330 participants with a national identification (ID) number that could be used to determine their survival status (Figure 1); participants were followed up for a median of 11.7 years until 31 January 2020. All participants provided written informed consent, and this study was approved by the Research Ethics Committee of the National Health Research Institute (EC1080701-E; Miaoli, Taiwan).

### 2.2. Frailty

Frailty and prefrailty were assessed using the FRAIL scale to assess frailty which was proposed by the International Academy on Nutrition and Aging [19]. It has been validated in previous studies [20]. The FRAIL scale includes five criteria: fatigue, resistance, ambulation, illnesses, and loss of weight [19]. Fatigue was measured by asking: “how much of the time during the past four weeks have participants felt tired?” The responses were measured from 1 (never) to 6 (all the time). Answers of 5 (most of the time) or 6 (all the time) were scored as 1; the other answers were scored as 0. Resistance was assessed by asking the participants how difficult it was to climb a flight of stairs. Responses of “Yes, limited a little or a lot” were scored as 1. Ambulation was assessed by asking the participants about the difficulty that they have walking a one block distance. Responses of “Yes, limited a lot” or “Yes, limited a little” were given a score of 1; the other responses were scored as 0. Illnesses were assessed by asking the participants if a medical professional had ever diagnosed them with any of the following illnesses: hypertension, cancer, diabetes, chronic lung disease, heart disease, asthma, arthritis, stroke, or kidney disease. A report of five or more illnesses received a score of 1. Weight loss was assessed by asking the participants if they had lost >5% of their weight in the past year. However, there was no equivalent variable that was available from the 2005–2008 NAHSIT database. Therefore, in this study, loss of weight was assessed by the participants who reported unintentional weight loss in the past month. Responses of “Yes” were given a score of 1. FRAIL scores ranged from 0 to 5, and these indicated frail (3–5), prefrail (1–2), and robust (0) conditions.

### 2.3. Dietary Intake Information

Dietary information was obtained by issuing the 24-h dietary recall and simplified food frequency questionnaire (FFQ). The 24-h dietary recall was used to estimate the nutrient intake, and the data were used to calculate the DDS [21]. The FFQ contains questions on the frequency of the consumption of 60 main food items per month/week/day during the month prior to taking the questionnaire. The validation of a similar simplified FFQ was previously reported. The Spearman rank correlation coefficients between the 24-h dietary recall and FFQ data ranged from 0.132 (whole grain) to 0.678 (dairy) for men, and −0.052 (whole grain) to 0.759 (dairy) for women [22]. 

Dietary quality was assessed using the DDS [10]. It was derived from the 24-h dietary recall of six food groups including dairy, soybeans/fish/eggs/meat, grain, fruit, vegetables, and oil/fat/nuts, in accordance with the Taiwanese Food Guides, and yielded a total score of 6 points. Achieving half of the recommended serving numbers per day of the six groups was required for a score of one [9]. The method that was used to estimate the serving numbers of the six food groups is provided in the Appendix A. A DDS of >4 and ≤4 were considered high and low dietary diversity [9], respectively.

### 2.4. Mortality

The National Death Registry data were obtained from the Health and Welfare Data Science Center, Ministry of Health and Welfare. The 2005–2008 NAHSIT dataset of 330 participants with valid identification was linked to the National Death Registry to determine the participants’ survival status; they were followed up until 31 January 2020.

### 2.5. Statistical Analysis

All data analyses were performed using SAS statistical software (version 9.4; SAS Institute, Cary, NC, USA). Descriptive statistics are presented as the means ± standard deviations (SDs) and percentages for continuous and categorical variables, respectively. We evaluated the distribution of the differences within demographic and frailty severity by employing the one-way analysis of variance and the chi-square test or Fisher’s exact test. We calculated the follow-up time from the date of the interview to either the date of their death or 31 January 2020. The Cox proportional hazards regression model was used to evaluate the association between frailty severity and all-cause mortality, and the additional analyses were in accordance with the DDS. However, the sample size was limited for frailty; thus, to ensure that there was statistical power, we combined the categories of prefrailty and frailty into “frailty status” in the joint effect analysis. 

For all covariates except for body mass index (BMI), less than 6% of the values were missing and all of them were imputed to the mode value (all covariates were categorical variables). The proportion of missing values was high for BMI (34.5%). To avoid the massive imputation for a non-negligible number of participants, the exclusion of those with missing data, or the introduction of a bias, we included a missing class into the models for this variable.

## 3. Results

The baseline characteristics of the participants are shown by their frailty severity as shown in Table 1. A total of 330 participants (180 men [54.5%] and 150 women [45.5%]) were included in this study, and the proportions of prefrailty and frailty were 36.1% and 5.8%, respectively. The distribution of frailty severity was similar to that of the original dataset (Appendix A). Frail older adults often had a lower education level, poorer perceived health and sleep quality, and a higher number of drug treatments. Those with prefrailty consumed a significantly higher proportion of dietary supplements. It is noteworthy that when they were compared to those features of men, frail, older women had a higher BMI, waist circumference, and hip circumference (all *p* < 0.05; Appendix A).

The DDS for robust, prefrail, and frail participants were 4.74 ± 0.97, 4.61 ± 1.10, and 4.32 ± 1.16, respectively, but it did not show a significant difference for frailty severity in this study (*p* = 0.069). In addition, the mean consumption frequencies of egg, dairy products, noodles products, breakfast cereals, vegetables, mushroom, fresh fruit, nuts and seeds, poultry/livestock blood and other parts, and coffee increase with the DDS score. Those with a lower DDS had consumed less dairy, vegetables and fruits (Appendix A).

During the median follow up of 11.7 years, 148 cases of death were recorded in the National Death Registry. Table 2 shows the associations between frailty severity and the risk of all-cause mortality. The crude model shows that frail, older people had a higher risk of mortality when it was compared with that of robust individuals (hazard ratio [HR]: 1.70, 95% confidence interval [CI]: 1.31–2.20). With adjustments for age and sex (Model 1), and additional adjustments for education level, smoking status, alcohol consumption, monetary status, sleep status, cognitive function, dietary supplement use, number of drugs that were being used (Model 2), DDS (Model 3), and perceived health status (Model 4), the HR was 3.31 (1.60–6.84).

We merged the categories of prefrailty and frailty into ‘frailty status’ in the analysis. The combined effects of frailty status and DDS (≤4 and >4) on all-cause mortality are presented in Table 3. In Model 1, older adults with a frailty status and a lower DDS had increased risks of mortality (HR: 2.01, 95% CI: 1.26–3.19) than those without frailty and a diverse diet. Further adjustments for Model 2 and Model 3 covariates did not modify these findings. Sensitivity analyses (adjusted for Model 1 covariates), excluding participants who died in the first year of the follow up, provided similar results (HR: 2.25, 95% CI: 1.41–3.59; *p* = 0.007 for mortality risk).

## 4. Discussion

In this prospective cohort study, we found that frailty which was based on the FRAIL scale definition can predict mortality among older Taiwanese people, and their survival may be improved by having dietary diversity. Older adults with a frailty status and a lower DDS (DDS ≤ 4) had a higher risk of mortality when it was compared with that of robust adults or those with a frail status and a higher DDS (DDS > 4).

The systematic review and meta-analysis showed that the FRAIL scale can effectively identify frailty severity and predict disability among community-dwelling middle-aged and older people [23]. However, mortality studies on frailty and its severity are still limited [24]. We found that the FRAIL scale can predict the risk of mortality, and these findings did not change after adjustments were made for several covariates. In addition, the prevalence of prefrailty and frailty was 35.8% and 3.9%, respectively, among NAHSIT 2005–2008 participants (Appendix A). Our previous study used representative data from the 2014–2016 NAHSIT of 1115 participants (aged ≥ 65 years), and found that the prevalence of prefrailty and frailty was 37.3% and 6.2%, respectively [11]. These results demonstrate that the prevalence of prefrailty and frailty is on the rise in Taiwan, and thus, it needs special attention.

We conducted the analysis by sex and found that frail, older women, but not men, had a higher BMI. These findings were consistent with a French longitudinal study of 1593 non-institutionalized older people that were aged ≥65 years. The results demonstrated that older women were approximately 2-fold more likely to be frail when they were obese, but there was no such correlation in men [25]. Moreover, a higher BMI was inversely associated with skeletal muscle mass, muscle strength, and bone mass, which may have been due to adipose tissue involvement in the complex bone-muscle interaction [26]. Therefore, older adults should maintain both a stable weight (BMI 24–26.9 kg/m^2^) and also skeletal muscle mass to prevent frailty and subsequent disability and mortality [27,28]. 

In a Japanese prospective cohort study of 666 community-dwelling older adults, a “sugar and fat” dietary pattern had a positive association with frailty [29]. The consumption of a high sugar or fat diet has been shown to increase mitochondrial dysfunction and inflammation [30]. It may increase proteolysis and reduce skeletal muscle protein synthesis, and further reduce muscle function and strength [31]. Frailty may be the result of tissue damage that is caused by a poor antioxidant ability [32]. It has been suggested that frail, older adults tend to have a poor appetite or anorexia [33]. These diets were characterized by reducing the intake of certain food groups (e.g., protein-rich food and fruit and vegetables), and these adults often have a lower DDS, which typically infers the consumption of higher amounts of sugar and fat (e.g., refined foods and fatty meats) [14,33,34]. These aging-related overall dietary quality changes may increase the risk of frailty [16,35] and mortality [14,36].

The consumption of a single food or supplement alone cannot improve frailty [37]. Studies have been shown that dietary diversity is inversely associated with frailty [11,38]. Also, dietary interventions have been shown to improve frailty severity, which when they are combined with resistance training, lead to a great improvement in frailty severity [39]. Antioxidant polysaccharides from food sources such as vegetables, fruits, cereals, beans, mushrooms, tea, milk products, and shellfish are associated with antioxidant function [40]. Moreover, in this study, older adults with a higher DDS demonstrated a higher consumption frequency of vegetables, fruit, cereals, dairy products, and mushrooms (Appendix A). Nevertheless, we did not find an association between dietary diversity and frailty severity in this cross-sectional study, but the DDS showed that there is a long-term protective effect. This study had some limitations. First, frailty and dietary data were collected at a baseline, which cannot assess the effects of the changes that occur or determine the causality of frailty. Second, we used one 24-h dietary recall test to measure the DDS, which may not completely capture the long-term DDS. Therefore, the composite responses from the FFQ were used to support the validity of the DDS from one 24-h dietary recall test in this study of the older adults. We have provided the distribution of food intake frequency as determined by the DDS in the Appendix A. Moreover, these estimates, even though there is less variation in them, still illustrate our point, and the effect that they have may have been underestimated. Third, only 330 people with a valid identification could be linked to mortality. The information about mortality could be biased. The strength of this study is that it was a cohort study, with comprehensive socio-demographic characteristics as well as dietary and nutritional information.

## 5. Conclusions

In this prospective study, older adults with frailty had an increased risk of mortality. Higher dietary diversity might reduce the risk of mortality in older adults with prefrailty and frailty. Dietary diversity has health benefits. Health promotion interventions should emphasize the importance of eating a variety of foods, especially for older adults who tend to have reduced food intakes, and especially for those with a lower DDS who consume less dairy, vegetables and fruits. Further studies are needed to examine this association and investigate whether combined dietary and exercise interventions can reduce frailty and mortality in older adults.

## Figures and Tables

**Figure 1 nutrients-14-03825-f001:**
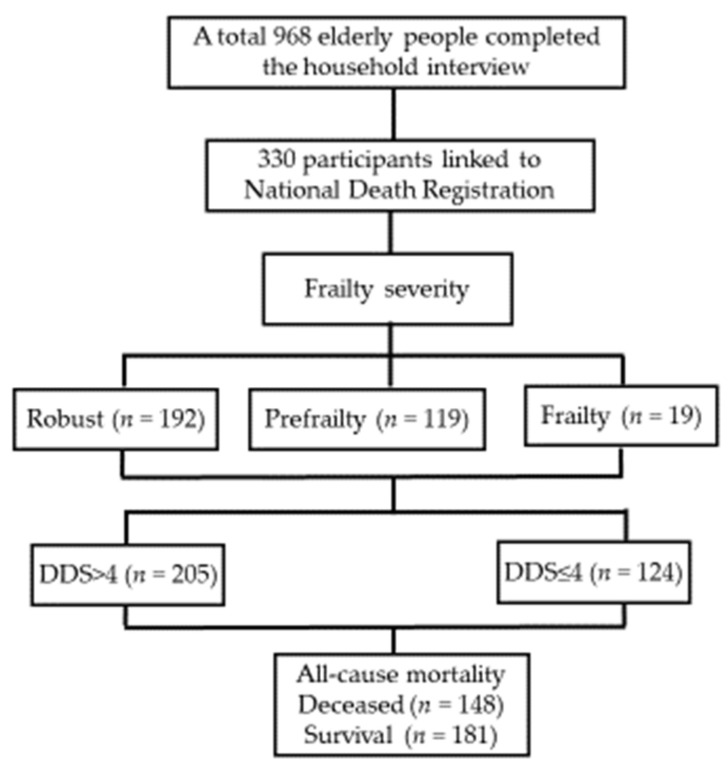
Flow chart for participant selection.

**Table 1 nutrients-14-03825-t001:** Baseline characteristics by frailty severity among NAHSIT 2005–2008 older adults (*n* = 330).

	Frailty Severity			
	Robust	Prefrailty	Frailty	P-Trend
*n*, %	192 (58.2)	119 (36.1)	19 (5.8)	
Gender, %				0.763
Men	56.3	52.1	52.6	
Women	43.8	47.9	47.4	
Age (years), %	71.8 ± 5.47	73.5 ± 2.20	73.7 ± 5.44	0.008
65–69	42.7	30.3	36.8	0.234
70–74	25.0	26.9	15.8	
75–79	22.9	29.4	26.3	
≥80	9.38	13.5	21.1	
Education, %				0.048
Illiterate	19.8	26.9	36.8	
Primary school	60.9	45.4	42.1	
High school and above	19.3	27.7	21.1	
Smoking, %	35.4	36.2	36.8	0.985
Drink alcohol, %	42.1	37.1	36.8	0.659
Enough money, %				0.811
More than enough	8.47	10.3	5.88	
Just enough	56.6	50.0	52.9	
difficult	34.9	36.7	41.2	
Perceived health status, %				<0.001
Good	34.2	24.4	0.00	
Fair	54.6	42.6	17.7	
Poor	11.2	33.0	82.4	
Poor sleep quality, %	7.3	17.2	26.3	0.006
Number of diseases	2.03 ± 1.23	3.86 ± 2.20	5.84 ± 1.68	<0.001
Number of medicines	1.03 ± 1.03	2.06 ± 1.36	3.63 ± 1.95	<0.001
Cognitive impairment^¶^, %	15.3	22.4	31.6	0.342
Supplement use, %	43.3	62.2	36.8	0.002
BMI (kg/m^2^)	24.4 ± 3.17	24.6 ± 3.52	26.7 ± 5.48	0.181
Waist circumference (cm)	85.3 ± 10.1	86.2 ± 10.2	91.5 ± 9.11	0.106
Hip circumference (cm)	92.6 ± 6.48	93.4 ± 6.68	96.8 ± 9.70	0.080
DDS (score)	4.74 ± 0.97	4.61 ± 1.10	4.32 ± 1.16	0.069

**Abbreviations**: NAHSIT, Nutrition and Health Survey in Taiwan; BMI, body mass index; DDS: dietary diversity score. Categorical variables are presented as a percentage, and continuous variables are presented as mean ± standard deviation (SD). General linear model regression and chi-square or Fisher’s exact tests were used for continuous and categorical variables, respectively. Cognitive impairment^¶^ was defined by a short, portable mental status questionnaire (SPMSQ) with ≥3 errors.

**Table 2 nutrients-14-03825-t002:** Association between frailty severity and risk of all-cause mortality in 2005–2008 NAHSIT older adults (*n* = 330).

	Frailty Severity			
	Robust	Prefrailty	Frailty	*p*-Value
All-cause mortality				
Deceased/survival (*n*)	75/117	58/61	15/4	
Crude	1.00	1.42 (1.01–2.00)	3.73 (2.13–6.52)	<0.001
Model 1	1.00	1.30 (0.92–1.85)	3.80 (2.15–6.71)	<0.001
Model 2	1.00	1.42 (0.98–2.07)	5.09 (2.63–9.84)	<0.001
Model 3	1.00	1.42 (0.98–2.07)	5.10 (2.63–9.87)	<0.001
Model 4	1.00	1.19 (0.80–1.78)	3.31 (1.60–6.84)	0.004

The Cox proportional hazards model was estimated for hazard ratios. Model 1: adjusted for age and sex. Model 2: Model 1 plus education level, smoking status, alcohol consumption, monetary status, sleep status, cognitive function, supplement use, and the number of drug treatments that were being used. Model 3: Model 2 plus DDS. Model 4: Model 3 plus BMI.

**Table 3 nutrients-14-03825-t003:** Association between frailty status (combined prefrailty and frailty) stratified by DDS and all-cause mortality in 2005–2008 NAHSIT older adults *(n* = 329).

	Frailty Severity				
	Robust		Prefrailty/Frailty		
	DDS > 4	DDS ≤ 4	DDS > 4	DDS ≤ 4	*p*-Value
All-cause mortality					
Deceased/survival	44/79	31/37	39/43	34/22	
Crude	1.00	1.27 (0.80–2.02)	1.50 (0.97–2.31)	2.25 (1.44–3.53)	0.004
Model 1	1.00	1.16 (0.72–1.85)	1.35 (0.88–2.09)	2.01 (1.26–3.19)	0.024
Model 2	1.00	1.05 (0.64–1.74)	1.56 (0.99–2.47)	1.79 (1.07–3.02)	0.062
Model 3	1.00	1.08 (0.54–2.19)	1.30 (0.72–3.74)	2.30 (1.11–4.75)	0.118

Cox proportional hazards model was estimated for hazard ratios. Model 1: adjusted for age and sex. Model 2: adjusted additionally for education level, smoking status, alcohol consumption, monetary status, sleep status, cognitive function, supplement use, and the number of drug treatments that were being used. Model 3: Model 2 plus BMI.

## Data Availability

Restrictions apply to the availability of these data. Data was obtained from Wen-Harn Pan, and are available Wen-Harn Pan with the permission of https://www.ibms.sinica.edu.tw/wen-harn-pan/ (accessed on 31 July 2022).

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
