# Peer review of "The Combined Effects of Dietary Diversity and Frailty on Mortality in Older Taiwanese People"

_nutrients, 2022, doi:10.3390/nu14183825_

Round 1
Reviewer 1 Report
Dear Authors,
Thank you for the opportunity to review your manuscript “The combined effects of dietary diversity and frailty on mortality in older Taiwanese”
Firstly, there is a lack of consistency in the objective and aim: The objective of this manuscript was to assess the prospective association between frailty and dietary diversity on mortality. Although this has been the objective, the aim is stated as “ to investigate the combined effects of dietary diversity and frailty severity on all-cause mortality in older adults” which although similar is not consistent
Line 34: you state: “complicated geriatric syndrome” will you please explain this terminology.
Line 68: “total of 968 older adults” why was this number used, was their a G Power analysis completed and what is the associated confidence level to assess and accept the results???
Line 97: What FFQ was used, and how was it validated.
Line 104: What is a serving, please relate this to gms/day so that a standardized assessment can be made.
Why is there no FFQ assessment with food groups shown in the results and discussed??
Line 189 : You use the term dietary diversity. Please explain what this means and how it relates to this population.
Line 212 you state: “Dietary sugar and fat are associated with oxidative stress and inflammatory processes” why is this important, how does this relate to the outcomes and the impact on frailty or death??
Line 218 you state” Consumption of a single beneficial food” what is a beneficial food and how has this been determined within the age group under review.
In your conclusion it would be helpful if you give an overview of the food groups of benefit and why, the exercise to be performed and compare with the frailty index and a possible suggestion of future dietary plans that would reduce the risk of frailty and or the incidence of death.
Reviewer 2 Report
Thank you for this manuscript in which you combined databases to obtain insight into associations between dietary diversity, frailty and mortality.
My comments:
- line 53: there seems to be words missing in this sentence?: 'with both..'?
- line 60: You state that previous research shows that dietary diversity might play an important role in affecting health related outcome. This is really not new. After this statement you say: "Therefore, the aim of this study was to investigate the combined effects of dietary diversity and frailty severity on all-cause mortality in older adults". In my opinion, the introduction section could be improved by explaining more on frailty. What is frailty and why would it be interesting to study diet in relation to frailty and the combined effects in relation to mortality? What is known already about this relationship?
- line 62: the research is about studying associations, not effects
Author Response
Please see the attachment.
- - line 53: there seems to be words missing in this sentence? 'with both..'?
Response: We have revised our Introduction: A Taiwanese cross-sectional study demonstrated that dietary pattern with more fruit, nuts and seeds, tea, vegetables, whole grains, shellfish, milk/yogurt/cheese, fish had a reduced prevalence of frailty (Lines 56-59). In Line 56: It is possible that the cause of frailty might not just be inadequate of individual nutrients or foods.
- - line 60: You state that previous research shows that dietary diversity might play an important role in affecting health related outcome. This is really not new. After this statement you say: "Therefore, the aim of this study was to investigate the combined effects of dietary diversity and frailty severity on all-cause mortality in older adults". In my opinion, the introduction section could be improved by explaining more on frailty. What is frailty and why would it be interesting to study diet in relation to frailty and the combined effects in relation to mortality? What is known already about this relationship?
Response: Higher dietary diversity has long-term health benefits, and had a lower risk of all-cause mortality (Kant et al. 1993)(Lee et al. 2011). Our previous study demonstrated that frailty severity was inversely associated with DDS (Huang et al. 2021). The prospective cohort study shown that a poor diet quality may increase the incidence of frailty. It is unknown whether dietary diversity can reduce mortality associated with frailty. Therefore, we hypothesized that older adults with frailty and higher DDS had a higher survival rate.
We have revised our Introduction: Frailty is a medical and complicated geriatric syndrome characterized by multisystem decline associated with decreased functional reserve and increase vulnerability with aging (Clegg et al. 2013). Most studies define physical frailty by unintentional weight loss, exhaustion, low physical activity, slowness, and low grip strength (Fried et al. 2001) (Line: 34-37).
A Taiwanese cross-sectional study demonstrated that dietary pattern with more fruit, nuts and seeds, tea, vegetables, whole grains, shellfish, milk/yogurt/cheese, fish had a reduced prevalence of frailty (Line: 56-59)
- - line 62: the research is about studying associations, not effects
Response: Thank you for reminding us. We have made the changes in the Introduction: Therefore, the aim of this study was to investigate the prospective associations of dietary diversity and frailty severity on all-cause mortality in older adults (Line 64).
Again, thank you for giving us the opportunity to strengthen our manuscript with your valuable comments and queries. We are hopeful that the Editors find this response adequate to allow a favorable decision about our paper.
References
Clegg, A., J. Young, S. Iliffe, M. O. Rikkert, and K. Rockwood. 2013. 'Frailty in elderly people', Lancet, 381: 752-62.
Fried, L. P., C. M. Tangen, J. Walston, A. B. Newman, C. Hirsch, J. Gottdiener, T. Seeman, R. Tracy, W. J. Kop, G. Burke, and M. A. McBurnie. 2001. 'Frailty in older adults: evidence for a phenotype', Journals of Gerontology. Series A: Biological Sciences and Medical Sciences, 56: M146-56.
Huang, W. C., Y. C. Huang, M. S. Lee, H. Y. Chang, and J. Y. Doong. 2021. 'Frailty Severity and Cognitive Impairment Associated with Dietary Diversity in Older Adults in Taiwan', Nutrients, 13.
Lee, M. S., Y. C. Huang, H. H. Su, M. Z. Lee, and M. L. Wahlqvist. 2011. 'A simple food quality index predicts mortality in elderly Taiwanese', The Journal of Nutrition, Health & Aging, 15: 815-21.

Round 2
Reviewer 1 Report
Dear Authors,
Thank you for taking the time to include the suggestions made and they have certainly improved the manuscript.
This is now a far more interesting manuscript.